# Effectiveness of Following Mediterranean Diet Recommendations in the Real World in the Incidence of Gestational Diabetes Mellitus (GDM) and Adverse Maternal-Foetal Outcomes: A Prospective, Universal, Interventional Study with a Single Group. The St Carlos Study

**DOI:** 10.3390/nu11061210

**Published:** 2019-05-28

**Authors:** Nuria García de la Torre, Carla Assaf-Balut, Inés Jiménez Varas, Laura del Valle, Alejandra Durán, Manuel Fuentes, Náyade del Prado, Elena Bordiú, Johanna Josefina Valerio, Miguel A. Herraiz, Nuria Izquierdo, Maria José Torrejón, Maria Angeles Cuadrado, Paz de Miguel, Cristina Familiar, Isabelle Runkle, Ana Barabash, Miguel A. Rubio, Alfonso L. Calle-Pascual

**Affiliations:** 1Endocrinology and Nutrition Department, Hospital Clínico Universitario San Carlos and Instituto de Investigación Sanitaria del Hospital Clínico San Carlos (IdISSC), 28040 Madrid, Spain; nurialobo@hotmail.com (N.G.d.l.T.); carlaassafbalut90@hotmail.co.uk (C.A.-B.); i.jimenez.varas@gmail.com (I.J.V.); lauradel_valle@hotmail.com (L.d.V.); aduranrh@hotmail.com (A.D.); elena.bordiu@salud.madrid.org (E.B.); valeriojohanna@gmail.com (J.J.V.); pazdemiguel@telefonica.net (P.d.M.); cristinafamiliarcasado@gmail.com (C.F.); irunkledelavega@gmail.com (I.R.); ana.barabash@gmail.com (A.B.); marubioh@gmail.com (M.A.R.); 2Centro de Investigación Biomédica en Red de Diabetes y Enfermedades Metabólicas Asociadas (CIBERDEM), 28040 Madrid, Spain; 3Facultad de Medicina, Universidad Complutense de Madrid, 28040 Madrid, Spain; maherraizm@gmail.com (M.A.H.); nuriaizquierdo4@gmail.com (N.I.); mariaangeles.cuadado@salud.madrid.org (M.A.C.); 4Preventive Medicine Department, Hospital Clínico Universitario San Carlos and Instituto de Investigación Sanitaria del Hospital Clínico San Carlos (IdISSC), 28040 Madrid, Spain; mfuentesferrer@gmail.com (M.F.); nayade.delprado@salud.madrid.org (N.d.P.); 5Gynecology and Obstetrics Department, Hospital Clínico Universitario San Carlos and Instituto de Investigación Sanitaria del Hospital Clínico San Carlos (IdISSC), 28040 Madrid, Spain; 6Clinical Laboratory Department, Hospital Clínico Universitario San Carlos and Instituto de Investigación Sanitaria del Hospital Clínico San Carlos (IdISSC), 28040 Madrid, Spain; mjosetorrejon@gmail.com

**Keywords:** pregnancy, nutrition, MedDiet, real world, gestational diabetes mellitus, maternofoetal outcomes

## Abstract

We reported that a Mediterranean Diet (MedDiet), supplemented with extra-virgin olive oil (EVOO) and pistachios, reduces GDM incidence and several other adverse outcomes. In order to assess its translational effects in the real world we evaluated the effect of MedDiet from 1st gestational visit in GDM rate compared with control (CG) and intervention (IG) groups from the previously referred trial. As secondary objective we also compared adverse perinatal outcomes between normoglycemic and diabetic women. This trial is a prospective, clinic-based, interventional study with a single group. 1066 eligible normoglycaemic women before 12 gestational weeks were assessed. 932 women (32.4 ± 5.2 years old, pre-gestational BMI 22.5 ± 3.5 kg/m^2^) received a motivational lifestyle interview with emphasis on daily consumption of EVOO and nuts, were followed-up and analysed. Binary regression analyses were used to examine the risk for each pregnancy outcome, pregnancy-induced hypertension, preeclampsia, gestational weight gain (GWG), caesarean-section, perineal trauma, preterm delivery, small (SGA) and large for gestational age (LGA), and Neonatal Intensive Care Unit admissions. GDM was diagnosed in 13.9%. This rate was significantly lower than the CG: RR 0.81 (0.73–0.93), *p* < 0.001 and no different from the IG: RR 0.96 (0.85–1.07), *p* = 0.468. GWG was lower in diabetic women (10.88 ± 6.46 vs. 12.30 ± 5.42 Kg; *p* = 0.013). Excessive weight gain (EWG) was also lower in GDM [RR 0.91 (0.86–0.96); *p* < 0.001] without a significant increase of insufficient weight gain. LGA were also lower (1 (0.8%) vs. 31 (3.9%); *p* < 0.05)), and SGA were similar (5 (3.8%) vs. 30 (3.7%)). LGA were associated to EWG (RR 1.61 (1.35–1.91), *p* < 0.001). Differences in other maternal-foetal outcomes were not found. In conclusions an early MedDiet nutritional intervention reduces GDM incidence and maternal-foetal adverse outcomes and should be universally applied as 1st line therapy. GDM might not be consider as a high risk pregnancy any longer.

## 1. Introduction

Gestational diabetes mellitus (GDM) is one of the most common complications in pregnancy. Women with GDM and their offspring are at increased risk of short- and long-term complications, including, for mothers later development of type 2 diabetes (T2DM) and cardiovascular disease (CVD), and for offsprin, increased lifelong risk of obesity, T2DM, and metabolic syndrome [1,2,3]. The Hyperglycaemia and Adverse Pregnancy Outcome (HAPO) study established a continuous relationship between maternal hyperglycaemia and adverse outcomes [1], and a one-step approach for the diagnosis of GDM using a 75-g 2-h oral glucose tolerance test (OGTT) was proposed [4]. Recently, the HAPO follow-up study has also shown a continuous association between maternal glucose levels in pregnancy and late childhood glucose and insulin resistance, independent of maternal and childhood body mass index (BMI) and family history of diabetes [5,6]. The new criteria have been adopted by several societies, including the American Diabetes Association, the Endocrine Society and WHO [7,8,9]. However, the use of new criteria has been associated with an increase in incidence from 10.6% with Carpenter Coustan criteria to 35.5% in our population of women [10]. This dramatic increase in GDM diagnosis, together with higher rates of obesity, sedentary lifestyle, and older age at pregnancy -all risks factors related to GDM- has become a growing health problem demanding preventive strategies.

The idea of preventing GDM with diet is very attractive. However, studies show conflicting results depending on the type of nutritional intervention and the moment of its implementation. In addition, most studies have included only women with high risk of developing GDM [11]. A Mediterranean Diet (MedDiet), reinforced by the use of extra virgin olive oil (EVOO) and nuts, has proven to be beneficial in preventing T2DM, CVD, and GDM [12,13,14]. Moreover, our group has shown in the “St. Carlos GDM prevention study” [14] that a universal and early (end of first trimester) nutritional intervention based on the supplementation of the MedDiet with EVOO and pistachios, reduces significantly not only the incidence of GDM in 30%, but several other adverse outcomes. In addition, fewer women diagnosed with GDM in the intervention group required insulin therapy and gestational weight gain (GWG) was lower in the intervention group. This intervention has proven to be beneficial in both, women with GDM [15] and women with normal glucose tolerance (NGT) [16]. We would, therefore, recommend the adoption of this nutritional intervention by all pregnant women, from the beginning of gestation, not only due to its beneficial effects during pregnancy and birth, but also, because potential implications on the long-term health of the mother and their offspring.

So far, MedDiet has proven to be beneficial in preventing T2DM, CVD and GDM in clinical trials that provided EVOO and nuts to ensure adherence to the recommendations. Nevertheless, it is indispensable to assess the translational effects of this intervention to the daily clinical setting. In this project we have hypothesized that an easy, universal nutritional intervention in pregnant women based on a Mediterranean style of diet rich in EVOO and nuts (without providing any of them), applied in the usual clinical setting as early as possible, with a motivational interview instead of an educational group session, is not different from the results obtained from the previously reported clinical trial [14]. In addition, we will assess whether perinatal adverse outcomes associated to GDM could be reduced to be similar to those from women with NGT, and therefore GDM might not be consider as a high risk pregnancy any longer.

## 2. Methods and Materials

Our primary objective is to evaluate in women with normal fasting blood glucose (FBG < 92 mg/dL (5.1 mmol/L)) at the first gestational visit (8–12 gestational weeks (GW)) the effect of Mediterranean-based nutritional recommendations in the real world in the incidence of GDM and assess its effectiveness compared with the control (GDM 23.4%) and intervention (GDM 17.1%) groups from the previous clinical trial [14]. Secondary outcome is to assess the effect of the dietary intervention on the percent of diabetic women requiring insulin therapy, gestational weight gain (GWG), pregnancy-induced hypertension, caesarean section, perineal trauma, preterm delivery (<37 GW), neonates SGA (<10 percentile) and LGA (>90 percentile) according to national charts, admissions to the Neonatal Intensive Care Unit (NICU), and UTIs.We will compare adverse perinatal outcomes in women with GDM and women with NGT, since an early nutritional intervention previous to GDM diagnosis could ameliorate differences between groups.

### 2.1. Study Design

This is a prospective, unicentric, clinic-based, interventional study with a single group. This trial was registered October the 11th 2016 with the number ISRCTN13389832 (DOI 10.1186/ISRCTN13389832).

### 2.2. Setting

The Hospital Clínico San Carlos in Madrid, Spain provides specialist health care to an estimated population of 370,000 people in Madrid, Spain. Universal screening for GDM is performed in all pregnant women.

### 2.3. Participants and Study Conduct

The Institutional Review Board and The Clinical Ethic Committee of the Hospital Clínico San Carlos have approved this protocol before initiation of the study (approval reference: 16/442-E). We have included pregnant women who fulfil the inclusion criteria and none of the exclusion criteria. Subjects must have given signed and dated informed consent prior to any trial-related activities.

*Inclusion criteria*: women >18 year old, with normal FBG (<92 mg/dL (5.1 mmol/L)) in the 1st gestational assessment (8–12 GWs) and who signed the informed consent.

*Exclusion criteria*: women with FBG ≥ 92 mg/dL (5.1 mmol/L) in the 1st gestational assessment (8–12 GWs), multiple pregnancy, nut allergy, or any other reason, medical condition, ongoing medication or significant disability that would prevent the participant complying with trial consent, treatment and follow-up procedures or potentially jeopardize her medical care.

All women attending their first gestational visit at 8–12 GW between November 2016–November 2017 with FBG < 92 mg/dL (5.1 mmol/L) were assessed for inclusion. They were invited to participate upon their first ultrasound visit, between 12–14 GW. Gestational age at entry for inclusion was based on the one obtained in this first ultrasound.

We recruited 1066 patients. Of these, 932 women (32.4 ± 5.2 years old and pre-gestational BMI 22.5 ± 3.5 kg/m^2^) underwent a 75 g OGTT for GDM diagnosis and were analysed. Table 1 shows basal characteristics of trial population.

Women were followed up by the Obstetric Department as usual care and they were referred to the Diabetes and Pregnancy Unit if GDM was diagnosed. Participants were enrolled at 8–12 GWs (visit 1) and were followed-up at GWs 24–28 (visit 2), 34–36 (visit 3) as well as at 12–14 weeks post-partum (visit 4) (Figure 1).

Last patient was included in November the 21st 2017, gave birth on May the 7th 2018 and had the post-partum assessment on July the 30th 2018. Women attending later than 14 weeks post-delivery were not included in the post-partum analysis given the influence of breast-feeding in glucose regulation and insulin sensitivity.

For the present study protocol we finalized recruitment in November 2017, but this is an ongoing evaluation and we will carry on with the epidemiological surveillance to follow-up GDM and perinatal complications rates. Addendum was updated in the trial record (4/12/2018). Learning from challenges and implementation of the intervention with time might lead to an improvement in outcomes.

### 2.4. Nutritional Intervention

Participants had an individual motivational lifestyle interview with a dietician one week after inclusion in visit 1. Nutritional intervention was based on the five basic recommendations of a Mediterranean style of diet: -two servings/day of vegetables, -three pieces of fruit a day—avoiding juices, even fresh, -exclusive use of EVOO for both raw consumption and cooking, and—consumption of a daily handful of nuts. Other recommendations were: three servings/day of skimmed dairy products, wholegrain cereals, two-three servings of legumes/week, moderate to high consumption of fish, a low consumption of red and processed meat, avoidance of refined grains, processed baked goods, pre-sliced bread, soft drinks, fast foods and precooked meals. Emphasis was put on the use of EVOO for cooking and dressings and the daily consumption of nuts. Women were explained the potential health benefits of following the dietary advice for themselves and offspring. In addition, women were given several meal and snack ideas where they could include all the different food options. We believe this made it easier and more appealing to adhere to the Mediterranean dietary pattern.

### 2.5. Lifestyle Evaluation

Nutrition intervention was evaluated using a semi-quantitative frequency questionnaire, based on the Diabetes Nutrition and Complications Trial (DNCT) study [17] and the 14-point Mediterranean Diet Adherence Screener (MEDAS) [18]. The DNCT questionnaire contains 15 items and evaluates general healthy eating habits. Three of the items consider physical activity (items 1–3) and 12 assess the food frequency intake (items 4–15). There are three options in the questionnaire: A, B and C. Option A (value +1) is associated with T2DM prevention while option C (value −1) is associated with increased risk. Therefore, A is the most favorable habit while C is the least favorable. Option B (value 0) is the intermediate between A and C, and is the minimum objective to be achieved. The Nutrition pattern is based on twelve questions. The score is between −12 and 12, and the objective is >5. The 14-points Mediterranean Diet Adherence Screener (MEDAS) was also used to obtain the MEDAS-derived PREDIMED score. The MEDAS questionnaire considers 14 items and the compliance of each item provides +1 points. During pregnancy the consumption of alcohol and juices were not evaluated as favorable, in the same way, consumption of raw vegetables was not required. Therefore, a score ≥ 7 was considered as ideal.

### 2.6. GDM Screening and Treatment

The diagnosis of GDM was established with a 75 g OGTT according to the criteria of IADPSG 2009 [4] and WHO 2013 [9]. Women diagnosed with GDM were referred to the Diabetes and Pregnancy Unit and treated according to local guidelines. A detailed description of the management protocol for GDM has been published elsewhere [15]. GDM was treated with insulin and/or diet, and therapy registered. One-week after diagnosis at the very latest, women had their first appointment at the Diabetes and Pregnancy Unit. Specific diet was not provided to women; they were advised just to carry on with the same recommendations based on MedDiet. To register glycaemic control, women were told to perform a six-point daily glycaemic profile, with fasting/pre-prandial and 1-h postprandial glycaemias. The goals were fasting and pre-prandial glucose < 90 mg/dL (5 mmol/L) and 1-h post-meal glucose < 120 mg/dL (6.6 mmol/L). Insulin therapy was initiated when > 50% of fasting or pre-prandial values were > 95 mg/dL (5.3 mmol/L) or 1-h postprandial levels were > 140 mg/dL (7.8 mmol/L). Insulin requirements were adjusted weekly, based on the 6-point daily glucose profile.

### 2.7. Clinical History

A family history of T2DM and metabolic syndrome when > 2 components were present in the same relative, obstetric history of GDM and miscarriages, educational status, employment, number of prior pregnancies, smoking habit, gestational age at entry were recorded at Visit 1.

### 2.8. Anthropometric Data

Pre-gestational body weight (BW) was self-referred and registered at Visit 1. BW in each visit was measured without shoes and with light-weight clothes. Weight gain was evaluated at 8–12, 24–28 and 36–38 GWs (in relation to pre-gestational BW at Visit 1). Blood pressure (BP) was measured with an adequate armlet when the participants had been seated for 10 min.

### 2.9. Biochemical Variables

Blood was drawn between 08.00 and 09.00 a.m., after an overnight fast. The following data were determined: HbA1c, standardized by the International Federation of Clinical Chemistry and Laboratory Medicine (IFCC); serum insulin; HOMA-insulin resistance (HOMA-IR), calculated as glucose (mmol/L) × insulin (mcUI/mL)/22.7; and FBG.

### 2.10. Maternal Outcomes

Gestational weight gain was referred to pre-gestational BW. Insufficient weight gain (IWG) was defined as a gestational weight gain 3 Kg below the designated target according to their pre-gestational BMI. Excessive weight gain (EWG) was defined as a gestational weight gain 3 Kg above the designated target according to their BMI. The objective for GWG (adequate weight gain, AWG) was: BMI < 20: > 15 kg, BMI 20–24.9: 12 (±3) kg, BMI 25–26.9: 9 (±3) kg, BMI 27–29.9: 6 (±3) kg, BMI 30–34.9: 3 (±3) kg, BMI > 35: 0 (±3) kg. Pregnancy-induced hypertension (≥140mmHg systolic BP/90 mmHg diastolic BP after 20 GW); preeclampsia (≥140mmHg systolic BP/90 mmHg diastolic BP with proteinuria ≥300 mg in 24-h after 20 GW; albuminuria (proteinuria ≥ 300 mg in 24-h with systolic BP <140 mmHg and diastolic BP < 90 mmHg); bacteriuria and UTIs (number of events requiring antibiotic treatment); and type of delivery (vaginal, instrumental or CS) and perineal trauma (any degree of spontaneous tears) and episiotomy, induction and epidural analgesia were recorded.

### 2.11. Neonatal Outcomes

Gestational age at birth, prematurity, birth weight (g), height (cm) and percentiles, LGA, SGA according to national charts, and NICU admissions, ph cord blood, Apgar score, hypoglycaemia (<40 mg/dL (<2.2 mmol/L) and < 30 mg/dL (1.7 mmol/L) in premature babies), hyperbilirubinemia, respiratory distress and brachial plexus injury were registered. Newborns of women with GDM had no specific indications of being admitted to NICU. These newborns were usually kept in neonatal nursing room for a 6 to 8 h period, independent to the NICU unless they required a NICU admission specifically for other reasons.

### 2.12. Sample Size

For sample size calculation, a primary outcome of GDM incidence was used, assuming an expected incidence of 16%, obtained from initials results from our previous study [14]. A sample size of 1514 women would estimate the incidence with a confidence interval (CI) of 95% and a precision of 2.5% and would allow us to differentiate the results from those obtained from the control group (23.4%) in the previous clinical trial [14]. After six months of recruitment we performed an intermediate revision of the protocol and obtained a GDM incidence of 14.5%, therefore we recalculated the sample size assuming an expected incidence of 14.5%. A sample size of 873 women would estimate the incidence with a CI of 95% and a precision of 2.5% and would allow us to differentiate the results from those obtained from the control group in the previous clinical trial [14] with a power of 97%.

### 2.13. Statistical Analysis

Categorical variables are presented with their frequency distribution. Continuous variables are given by their mean and standard deviation (±SD). Study of data distribution for continuous variables was performed through the Kolmogorov Smirnov test. Comparison between GDM group and NGT group characteristics for categorical variables was evaluated by the *χ*^2^ test or ANOVA. For continuous variables, measures were compared with Student’s t test or the Mann–Whitney *U* test if distribution of quantitative variables was or not normal. The differences of mean values between groups (GDM vs. NGT) for each analysed variable are given with 95% confidence interval (CI).

Logistic binary regression analyses were used to assess the effect of the GDM on adverse maternal and neonatal outcomes estimating RR and its 95% confidence intervals. Two-tailed p values were calculated, with *p* < 0.05 indicating statistical significance. Analyses were performed using SPSS, version 21 (SPSS, Chicago, IL, USA).

## 3. Results

GDM was diagnosed in 130 women (13.9%). This incidence was significantly lower than in the control group: RR 0.81 (0.73–0.93), *p* < 0.001 and no different from the intervention group: RR 0.96 (0.85–1.07), *p* = 0.468 from the St. Carlos GDM prevention study. 

One hundred (72.8%) were controlled only with diet and 30 (27.2%) required insulin treatment. Table 2 shows maternal clinical, biochemical and anthropometric data at 24–28 and 36–38 GW.

HbA1c was significantly higher in women with GDM at 24–28 GW (5.0 ± 0.3 vs. 4.9 ± 0.3, *p* < 0.0001), but no different from normoglycaemic women at 36–36 GW (5.2 ± 0.3 vs. 5.2 ± 0.3). Equally, fasting serum insulin (mcUI/mL) was significantly higher in women with GDM at 24-28 GW (10.0 ± 6.5 vs. 7.9 ± 5.3, *p* = 0.003), but no different from women with NGT at 36–36 GW (13.3 ± 19.6 vs. 12.0 ± 16.1).

MEDAS and Nutrition scores were no significantly different between groups at 24–28 GW but were significantly higher in diabetic women at 36–38 GW (7.7 ± 2.2 vs. 5.4 ± 1.8, *p* < 0.001 for MEDAS and 7.6 ± 3.2 vs. 3.2 ± 3.5, *p* < 0.001 for Nutrition score).

Weight gain from pre-gestation was no different between groups at 24–28 GW, but was lower in diabetic women at 36–38 GW (10.88 kg ± 6.46 vs. 12.30 ± 5.42, *p* = 0.013). Distribution of GWG was significantly different between groups independently of pre-gestational BMI. More diabetic women had IWG than women with NGT (26.9% vs. 16.8%, *p* < 0.001), and less diabetic women had EWG than normoglycaemic women (26.2% vs. 46.3%, (ANOVA *p* < 0.001)). Table 3 displays maternal-foetal outcomes.

GDM was associated with a lesser EWG as compared to AWG, (RR 0.91 (0.86–0.96), *p* < 0.001). Differences in other maternal outcomes were no found. Despite a lower birth weight (3.126 ± 465 g vs. 3.273 ± 468 g, *p* < 0.002), SGA rate was similar (5 (3.8%) vs. 30 (3.7%), *p* = 0.553) and LGA rate was significantly lower (1 (0.8%) vs. 31 (3.9%), *p* < 0.048). EWG was associated with an increased risk for LGA (RR 1.61 (1.35–1.91)). Differences in other neonatal outcomes were no found.

In the post-partum visit we only reported data from women who attended just between 12 and 14 weeks after delivery. We analysed 384 women (70 with GDM and 314 with NGT). All analysed women were breast-feeding. Known risk factors for GDM were higher in diabetic women: they were older (35.2 ± 4.3 vs. 32.7 ± 4.9, *p* < 0.001) and had a higher BMI pre-pregnancy (23.3 ± 4.2 vs. 22.4 ± 3.2, *p* = 0.036). HbA1c levels in the first visit post-partum were also significantly higher in women with previous GDM (5.3 ± 0.3 vs. 5.2 ± 0.3, *p* < 0.001). Adherence to nutritional recommendations according to the Nutrition questionnaire, remained higher after delivery in all women compared to pre-intervention scores, but lower that at the end of pregnancy. In addition, women with GDM maintained a higher adherence post-partum than women with NGT (5.2 ± 3.4 vs. 3.4 ± 3.8, *p* = 0.002) (Table 4).

## 4. Discussion

We recently demonstrated that a MedDiet supplemented with EVOO and pistachios reduces GDM rate from 23.4% to 17.1% [14]. The current study shows the translational effects of this intervention and demonstrates that perinatal adverse outcomes associated to GDM could be reduced to be similar to those from women with NGT. GDM was diagnosed in 130 women (13.9%). This incidence was significantly lower than in the control group (RR 0.81 (0.727–9.28), *p* < 0.001) and no different from the intervention group (RR 0.96 (0.85–1.07), *p* = 0.468] from the St. Carlos GDM prevention study [14]. A GDM rate similar to the intervention group was achieved with the usual follow-up without providing EVOO and pistachios.

Increasing incidence of GDM reflecting the rising prevalence of obesity and older age among women of childbearing age, demands measures that prevent or reduce the risk of developing GDM. Several approaches have been studied to evaluate the effect of lifestyle interventions on the onset of GDM in women with risks factors [11]. In women with no defined risks factors large population studies are absent. Only two trials prior to ours [19,20], have studied the potential benefits of dietary modulation in improving GDM risk in a randomly selected cohort. In the “Low-GI Diet in Pregnancy” study, 62 pregnant women were randomly assigned to either a low glycaemic index diet or to a high fibre, moderate-to-high glycaemic index diet. No differences were found in either GDM incidence or foetal birth weight [19]. A combined approach to diet and physical activity in minimizing GWG and preventing GDM has been evaluated in another randomized controlled trial with 100 pregnant women. Although the intervention consisting of focused dietary counselling, encouragement to increase physical activity and advice regarding appropriate weight gain, was associated with significantly less GWG, no differences were found in the incidence of GDM [20]. The present study was specifically designed to assess the potential benefits of a Mediterranean-based dietary recommendation in reducing GDM and improving perinatal outcomes in a non-selected population with non-defined risk factors. Results showed that a MedDiet recommendation in the real word is able to reduce GDM. In addition, when we intervene as early as possible in pregnancy, prior to GDM diagnosis and before 12 GW, to guarantee at least 3 months of nutritional treatment, adverse perinatal outcomes associated to GDM such as pregnancy-induced hypertension, CS, perineal trauma, prematurity, SGA and LGA, admissions to NICU and UTIs were no different between diabetic patients and women with NGT, and therefore GDM might not be consider as a high risk pregnancy any longer.

Birth weight was lower in new-borns from diabetic women due to a decrease in LGA without increasing SGA. Since there is a continuous relationship between maternal glycaemia and birth weight [1], these data indicate that some women even with glucose levels below those diagnostic of GDM with IADPS criteria could still be at risk. On the other hand, EWG was globally related with a higher risk of having LGA newborns. It has been shown that GDM treatment reduces obesity-induced adverse pregnancy outcomes including birth weight >90^th^-centile [21]. There was a lower GWG in diabetic patients at 36–38 GW but not at 24–28 GW, possibly due to reinforcement in the intervention after GDM followed by a higher adherence to nutritional recommendations. Diagnosis of GDM itself could have a direct impact on adherence to lifestyle counselling. Distribution of GWG was significantly different between groups and we found that GDM treatment significantly reduced EWG without significantly increasing IWG.

There was a higher adherence to Mediterranean diet assessed by MEDAS and Nutrition scores in diabetic women at 36–38 GW, but not at 24–28 GW, since nutritional intervention was reinforced once GDM had been diagnosed. Even in post-partum, women with GDM diagnosis maintained a higher adherence than women with NGT, but lower that at the end of pregnancy. The MedDiet’s protective effect is associated to a high intake of poly and mono-unsaturated fatty acids readily available in the Mediterranean diet as opposed to a Western diet rich in saturated fats [22]. EVOO is a rich source of mono-unsaturated fatty acids, and has been found to lower postprandial glucose levels [23] and to improve the inflammatory profile [24]. Furthermore, its liberal use might facilitate an increased intake of vegetables, traditionally eaten with olive oil in Spanish cuisine. Nuts are rich in unsaturated fatty acids and other bioactive compounds: high-quality vegetable protein, fiber, minerals, tocopherols, phytosterols, and phenolic compounds [25]. Nuts are able to improve glycaemic control in patients with T2DM. Diets emphasizing tree nuts intake have shown to significantly reduce HbA1c and FBG compared with isocaloric control diets in a pooled analysis of 450 patients from 12 trials [26]. We have previously shown a linear association between adherence at the end of the first trimester to the five basic recommendations of a Mediterranean style of diet and a lower risk of GDM, a composite of maternal-foetal outcomes (including emergency CS, perineal trauma, pregnancy-induced hypertension and preeclampsia, prematurity, LGA, SGA), UTIs, prematurity, and SGA new-borns [27].

“Pregnancy complications such as GDM, prematurity, UTI, new-born birthweight (large- and small-for- gestational- age) and pregnancy-induced hypertensive disorders have been associated with maternal diet. All these complications seem to be associated with proinflammatory processes and/or impaired glycaemic control. Dietary patterns rich in vegetables, fruits, fibre, fish, legumes, whole-wheat cereals and vegetable oil have shown to favour these complications [28,29]. The anti-inflammatory and antioxidant properties of this diet, along with its shown effects on improved insulin sensitivity could explain these results”.

The design of the trial has some limitations. The main one being that patients in this study are compared with two cohorts from a previous trial [14], and therefore we compare consecutive rather than parallel groups. It is possible that knowledge of the benefits of the previous study may motivate some women to adhere better to the recommendations given and staff to be more enthusiastic on counselling. Nevertheless, this is not a limitation as such, since our aim is to achieve a good adherence to MedDiet only through nutritional counselling in usual clinical practice. Assessing adherence to the intervention is crucial to accurately analyse the results. We used the Nutrition and MEDAS-derived PREDIMED scores. The original PREDIMED questionnaire considers moderate alcohol intake and juice consumption beneficial. However, pregnant women were advised not to consume either. In addition, consumption of raw vegetables was not required. Therefore, we did not expect patients to reach a score of ≥ 10 and a score ≥ 7 was considered as ideal. A better assessment of the dietary intake could be achieved by using specific biomarkers such as hydoxytyrosol and α-linolenic acid. Nevertheless, this is an expensive tool that is not easily applicable in regular clinical practice and it does not represent the implementation of MedDiet in the real world. Nevertheless, in the previous study [14] we found a good correlation between both parameters.

On summary, based on the translation of evidence to clinical practice we recommend that all pregnant women are encouraged to follow this protocol from early in pregnancy to reduce the burden of GDM.

## Figures and Tables

**Figure 1 nutrients-11-01210-f001:**
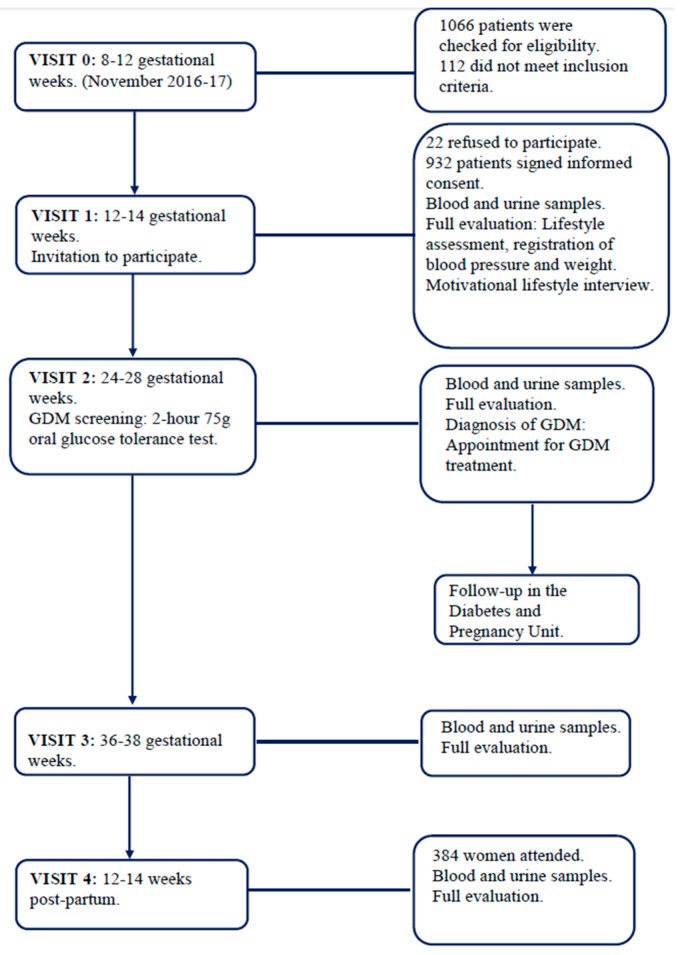
Flow chart of the study conduct.

**Table 1 nutrients-11-01210-t001:** Characteristics of the clinical trial population at baseline.

Variables	Mean ± SD or n (%)
N	932
Age (years)	32.4 ± 5.2
Race/Ethnicity	
Caucasian	612 (66.7)
Hispanic	284 (30.5)
Others	26 (2.8)
Family history of Type 2 Diabetes	260 (27.9)
Family history of MetS (>2 components)	174 (18.7)
Previous history of Gestational Diabetes	37 (4.0)
Previous history of Miscarriages	325 (34.9)
Educational status	
Elementary education	58 (6.2)
Secondary School	241 (25.9)
University Degree	621 (66.7)
UNK	12 (1.3)
Employment	743 (79.7)
Number of pregnancies	
Primiparous	394 (42.3)
Second pregnancy	276 (29.6)
>2 pregnancies	262 (28.1)
Never Smoker	531 (57.0)
Current Smoker	77 (8.3)
Gestational Age at baseline (weeks)	12.1 ± 0.5
Prepregnancy Body Weight (Kg)	59.6 ± 9.7
At entry Body Weight (Kg)	61.4 ± 9.9
Weight gain (Kg)	1.8 ± 3.4
Prepregnancy BMI (kg/m^2^)	22.5 ± 3.5
At baseline BMI (kg/m^2^)	23.4 ± 3.7
Systolic Blood Pressure (mm Hg)	108 ± 10
Diastolic Blood Pressure (mm Hg)	67 ± 8
Fasting Blood Glucose (mg/dL)	80 ± 6
mmol/dL	4.4 ± 0.3
HbA1c %	5.1 ± 0.2
mmol/mol	32 ± 0.8
Cholesterol mg/dL	169 ± 32
mmol/L	4.39 ± 0.83
Triglycerides mg/dl	79 ± 34
mmol/L	0.9 ± 0.4
TSH mcUI/mL	2.1 ± 2.9
T4L ng/dL	8.8 ± 1.6
MEDAS Score	4.3 ± 1.7
Nutrition Score	0.4 ± 3.1

Data are Mean ± SD or number (%) MetS, Metabolic Syndrome. BMI, body mass index; MEDAS Score: 14-point Mediterranean Diet Score.

**Table 2 nutrients-11-01210-t002:** Maternal clinical, biochemical and anthropometric data at 24–28 and 36–38 gestational weeks (GWs).

Variables	NGT Group (*n* = 802)	GDM Group (*n* = 130), (13.9%)	Differences (95% CI)	*p*
75 g-OGTT 24–28 GW				
FBG mg/dL mmol/dL	82.9 ± 4.9 4.6 ± 0.3	92.4 ± 8.6 5.2 ± 0.4	9.5 (8.4;10.5) 0.56 (0.51;0.61)	0.0001
1 h Blood Glucose mg/dL mmol/dL	118.3 ± 26.4 6.6 ± 1.5	164.1 ± 33.5 8.9 ± 2.0	45.7 (39.4;52.1) 2.3 (2.0;2.6)	0.0001
2 h Blood Glucose mg/dL mmol/dL	101.3 ± 19.8 5.6 ± 1.1	137.6 ± 31.6 7.6 ± 1.8	36.2 (31.3;41.2) 1.9 (1.7;2.1)	0.0001
HbA1c % (mmol/mol) 24–28 GW	4.9 ± 0.3 (30± 0.9)	5.0 ± 0.3 (31± 0.9)	0.15 (0.10;0.20)	0.0001
HbA1c % (mmol/mol) 36–38 GW	5.2 ± 0.3 (33± 1)	5.2 ± 0.3 (34± 1)	0.07 (0.00;0.14)	0.051
FBG 36–38 GW mg/dL mmol/dL	77.6 ± 10.4 4.2 ± 0.4	80.1 ± 10.0 4.4 ± 0.4	2.5 (0.3;4.7) 0.2 (0.1;0.3)	0.024
Fasting Serum Insulin mcUI/mL				
24–28 GW	7.9 ± 5.3	10.0 ± 6.5	2.1 (0.7;3.5)	0.003
36–38 GW	13.3 ± 19.6	12.0 ± 16.1	−1.3 (−5.7;3.2)	0.182
HOMA-IR				
24–28 GW	1.9 ± 2.7	2.3 ± 1.5	0.4 (−0.2;1.1)	0.558
36–38 GW	3.1 ± 5.5	2.7 ± 4.6	−0.4 (−1.6;0.9)	0.231
Cholesterol mg/dL (mmol/dL)				
24–28 GW	243 ± 42 (6.32 ± 1.09)	238 ± 43 (6.19 ± 1.12)	−4.7 (−12.9;3.5)	0.952
36–38 GW	272 ± 49 (7.07 ± 1.27)	261 ± 50 (6.79 ± 1.30)	−10.3 (−20.8;0.2)	0.054
Triglycerides mg/dL (mmol/dL)				
24–28 GW	155 ± 52 (1.77 ± 0.59)	167 ± 57 (1.90 ± 0.65)	11.8 (0.7;23.1)	0.038
36–38 GW	228 ± 74 (2.60 ± 0.84)	223 ± 69 (2.54 ± 0.79)	−4.5 (−20.5;11.5)	0.577
TSH (mcUI/mL)				
24–28 GW	2.0 ± 1.0	1.9 ± 1.3	−0.1 (−0.4;0.1)	0.215
36–38 GW	1.6 ± 1.0	1.7 ± 1.0	0.1 (−0.2;0.3)	0.602
T4L(ng/dL)				
24–28 GW	7.0 ± 1.2	7.0 ± 1.3	−0.0 (−0.2;0.2)	0.999
36–38 GW	7.1 ± 1.3	7.2 ± 1.3	0.2 (−0.1;0.5)	0.188
Treatment of GDM				
Nutritional		100 (72.8)		
Insulin (total)		30 (27.2)		
Bolus		5 (14.9)		
Basal		18 (74.5)		
Basal/Bolus		7 (10.6)		
Body Weight (kg)				
24–28 GW	66.5 ± 10.0	66.3 ± 10.3	−0.2 (−2.1;1.7)	0.807
36–38 GW	71.4 ± 9.9	70.3 ± 11.3	−1.1 (−3.1;0.9)	0.281
Weight gain (kg) pregestation to 24–28 GW	7.02 ± 4.27	7.13 ± 4.91	0.11 (−0.72;0.94)	0.804
Weight gain (kg) pregestation to 36v38 GW	12.30 ± 5.42	10.88 ± 6.46	−1.42 (−2.54; −0.30)	0.013
EWG	371 (46.3)	34 (26.2)		
AWG	296 (36.9)	61 (46.9)		
IWG	135 (16.8)	35 (26.9)		0.001
Systolic BP (mm Hg) 24–28 GW	105 ± 11	106 ± 11	1.8 (0.3;3.4)	0.098
Diastolic BP (mm Hg) 24–28 GW	63 ± 8	65 ± 9	1.6 (−0.5;3.7)	0.819
Systolic BP (mm Hg) 36–38 GW	115 ± 14	114 ± 11	−1.1 (−3.0;0.8)	0.661
Diastolic BP (mm Hg) 36–38 GW	71 ± 10	70 ± 10	−0.8 (−3.4;1.8)	0.019
MEDAs SCORE				
24–28 GW	5.2 + 1.8	5.4 + 1.6	0.2 (−0.2;0.7)	0.292
36–38 GW	5.4 + 1.8	7.7 + 2.2	2.2 (1.6;2.9)	0.001
NUTRITION SCORE				
24–28 GW	2.4 + 3.5	2.8 + 3.5	0.4 (−0.3;1.1)	0.257
36–38 GW	3.2 + 3.5	7.6 + 3.2	4.5 (3.6;5.4)	0.001

Data are Mean ± SD or number (%). Abbreviation: NGT, normal glucose tolerance; GDM, Gestational Diabetes Mellitus; OGTT, oral glucose tolerance test; FBG; fasting blood glucose; GW, gestational weeks; EWG, excessive weight gain; AWC, adequate weight gain; IWG, insufficient weight gain; BP, Blood Pressure.

**Table 3 nutrients-11-01210-t003:** Maternal and neonatal adverse outcomes.

	NGT Group (*n* = 802)	GDM Group (*n* = 130)	*p*	Crude RR (95% CI)
*Maternal outcomes*				
IWG/AWG	135/296 (31.3)	35/61 (36.5)	0.196	1.21 (0.83–1.75)
EWG/AWG	371/296 (55.6)	34/61 (35.8)	0.001	0.91 (0.86–0.96)
Pregnancy-induced hypertension	15 (1.9)	3 (2.3)	0.470	1.20 (0.42–3.41)
Preeclampsia	9 (1.1)	1 (0.8)	0.583	0.72 (0.11–4.62)
Albuminuria	4 (0.5)	2 (1.5)	0.199	2.41 (0.77–7.56)
Bacteriuria	120 (15.0)	35 (26.9)	0.001	1.85 (1.31–2.61)
Urinary Tract Infection	31 (3.9)	7 (5.4)	0.271	1.34 (0.67–2.67)
Delivery				
Vaginal	503 (62.7)	76 (58.5)	0.140	0.86 (0.43–1.71)
Instrumental	129 (16.1)	30 (23.1)		
Cesarean section	170 (21.2)	24 (18.5)		
Emergency-CS	22 (13.5)	3 (13.6)	0.739	0.84 (0.52–1.35)
Induction	490 (61.1)	78 (60.0)	0.442	0.96 (0.69–1.33)
Analgesia	661 (82.4)	106 (81.5)	0.445	0.95 (0.63–1.42)
Episiotomy	256 (31.9)	43 (33.1)	0.443	1.05 (0.75–1.47)
Perineal Trauma	265 (33.0)	36 (27.7)	0.133	0.80 (0.56–1.15)
Dystocia	10 (1.2)	2 (1.5)	0.516	1.20 (0.34–4.29)
*Neonatal outcomes*				
Gestational Age at birth (weeks)	39.5 ± 2.0	39.5 ± 1.7	0.716	
<37 GW	46 (5.7)	7 (5.4)	0.535	0.94 (0.46–1.92)
Birthweight (g)	3273 ± 468	3126 ± 465	0.002	
Percentile	50.0 ± 30.3	42.7 ± 28.3	0.005	
Length (cm)	49.4 ± 2.1	49.1 ± 1.9	0.202	
Percentile	42.7 ± 27.4	38.7 ± 26.6	0.263	
LGA > 90 percentile	31 (3.9)	1 (0.8)	0.048	0.21 (0.03–1.51)
>4500 g	3 (0.4)	1 (0.8)	NA	
SGA < 10 percentile	30 (3.7)	5 (3.8)	0.553	0.99 (0.87–1.14)
Ph Cord Blood	7.28 ± 0.13	7.27 ± 0.08	0.405	
<7.1	19 (2.4)	6 (4.6)	0.122	1.76 (0.86–3.59)
Apgar Score at 1min	8.8 ± 1.0	8.8 ± 0.8	0.846	
<5	11 (1.4)	1 (0.8)	0.484	0.59 (0.09–3.91)
Apgar Score at 5 min	9.8 ± 0.7	9.9 ± 0.3	0.196	
<7	5 (0.6)	0 (0)	0.471	NA
Hypoglycemia	7 (0.9)	1 (0.8)	0.691	0.89 (0.14–5.64)
Respiratorydistress	8 (1.0)	0 (0)	0.299	NA
Hyperbilurrubinemia	14 (1.7)	1 (0.8)	0.358	0.47 (0.07–3.17)
Brachial plexus	1 (0.1)	0 (0)	0.861	NA
NICU/observation	9(1.1)/17 (2.1)	0 (0)/2 (1.5)	0.257/0.492	NA/0.75 (0.20-2.81)

Data are Mean ± SD or number (%). Abbreviation: NGT, normal glucose tolerance; GDM, Gestational Diabetes Mellitus; RR, relative risk; CI, confidence interval; IWG, insufficient weight gain; CS, C-section; GW, gestational weeks; LGA, large-for-gestational-age. SGA, small-for-gestational-age. NICU, Neonatal intensive care unit. N.A. no applicable.

**Table 4 nutrients-11-01210-t004:** Clinical and laboratory 12–14 WEEKS post-delivery data.

	ALL	NGT	GDM	*p*
*N*	384	314	70	
Age (year)	33.2 ± 4.9	32.7 ± 4.9	35.2 ± 4.3	0.001
Pregravid BW (Kg)	59.8 ± 9.3	59.5 ± 8.8	60.9 ± 11.2	0.265
PG-BMI (Kg/m^2^)	22.6 ± 3.4	22.4 ± 3.2	23.3 ± 4.2	0.036
Gestational Weight Gain	12.06 ± 5.42	12.6 ± 5.2	10.0 ± 5.7	0.001
Postdelivery BW (Kg)	64.1 ± 10.4	64.2 ± 17.4	63.6 ± 9.2	0.702
PD-BMI (Kg/m^2^)	24.2 ± 4.1	24.2 ± 4.0	24.6 ± 4.1	0.518
BWPostD-Pregravid (Kg)	4.82 ± 5.66	5.2 ± 5.7	3.1 ± 5.1	0.022
WC (cm)	84.0 ± 9.0	83.9 ± 9.1	84.8 ± 8.6	0.545
>89.5 cm	71 (18.5%)	54 (17.2%)	17 (24.3%)	0.119
FP Glucose (mg/dL) mmol/dL	84.6 ± 7.6. 4.7 ± 0.4	84.3 + 7.6 4.7 ± 0.4	86.0 + 7.6 4.8 ± 0.4	0.093
>100 mg/dL	8 (2.1%)	6 (1.9%)	2 (2.9%)	0.443
FP insulin (mcUI/mL)	6.4 ± 5.3	6.6 ± 5.6	5.6 ± 4.0	0.172
HOMA-IR	1.6 ± 1.4	1.6 ± 1.3	1.6 ± 1.6	0.892
>3.5	20 (5.2%)	17 (5.4%)	3 (4.3%)	0.534
SBP (mm Hg)	111 ± 13	111 ± 13	111 ± 12	0.923
DBP (mm Hg)	73 ± 10	73 ± 11	74 ± 10	0.501
T-Cholesterol mg/dL mmol/L	196 ± 39 5.11 ± 1.01	195± 38 5.07 ± 0.99	205 ± 42 5.33 ± 1.09	0.055
HDL-C mg/dL mmol/L	64 ± 15 1.66 ± 0.39	63 ± 15 1.64 ± 0.39	68 ± 13 1.77 ± 0.34	0.199
LDL-C mg/dL mmol/L	122 ± 35 3.17 ± 0.91	120 ± 32 3.12 ± 0.83	130 ± 44 3.38 ± 1.14	0.314
Triglycerides mg/dL mmol/L	80 ± 44 0.91 ± 0.50	80 ± 42 0.91 ± 0.48	80 ± 52 0.91 ± 0.59	0.981
Apolipoprotein B (mg/dL)	91 ± 28	89 ± 28	97 ± 28	0.342
CPR (mg/dL)	2.0 ± 4.1	2.3 ± 4.1	1.2 ± 4.0	0.390
Albumin/creatinine ratio (mg/g)	11 ± 22	12 ± 24	7 ± 8	0.462
HbA1c-IFCC % (mmol/mol)	5.2 ± 0.3 33 ± 3	5.2 ± 0.3 33 ± 3	5.3 ± 0.3 34 ± 3	0.001
>5.7%	17 (4.4%)	10 (3.2%)	7 (10%)	0.034
TSH mcUI/mL	2.2 ± 4.6	2.2 ± 5.0	2.1 ± 1.8	0.975
FT4 (ng/dL)	8.3 ± 3.0	8.3 ± 3.3	8.0 ± 1.2	0.409
Nutrition Score				
Pregestational	0.8 ± 2.9	0.8 ± 2.9	0.8 ± 3.0	0.976
36–38 GW	4.8 ± 3.4	3.9 ± 3.0	7.8 ± 2.8	0.001
PostDelivery	3.7 ± 3.8	3.4 ± 3.8	5.2 ± 3.4	0.002
MedDiet Score				
Pregestational	4.5 ± 1.6	4.5 ± 1.6	4.5 ± 1.4	0.830
36–38 GW	6.0 ± 2.1	5.5 ± 1.8	7.9 ± 2.2	0.001
PostDelivery	4.5 ± 3.2	4.5 ± 3.1	4.7 ± 3.6	0.625

Data are Mean ± SDM or number (%). NGT, Normal glucose tolerance. GDM, Gestational Diabetes Mellitus. WC, waist circumference; BMI, Body mass index; BW, Body Weight; FP, fasting plasma; sBP, systolic blood pressure; dBP, diastolic blood pressure. PG, Pregravid; PostD, Postdelivery.

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
