# Peer review of "Effectiveness of Following Mediterranean Diet Recommendations in the Real World in the Incidence of Gestational Diabetes Mellitus (GDM) and Adverse Maternal-Foetal Outcomes: A Prospective, Universal, Interventional Study with a Single Group. The St Carlos Study"

_nutrients, 2019, doi:10.3390/nu11061210_

Round 1

Reviewer 1 Report

Dear Author

Thank you for the opportunity to review your manuscript “Effectiveness of following Mediterranean diet recommendations in the real world in the incidence of gestational diabetes mellitus (GDM) and adverse maternal-foetal outcomes: a prospective, universal, interventional study with a single group. The St Carlos Study.”

Although a very interesting outcome, I offer the following for your consideration.

You have stated that this is a Mediterranean Diet, however, your nutritional intervention does not indicate this ;

“Participants had an individual motivational lifestyle interview with a dietician one week after

inclusion in visit 1. Nutritional intervention was based on the five basic MedDiet recommendations:

 - two servings/day of vegetables, - three pieces of fruit a day - avoiding juices, even fresh, - exclusive

use of EVOO for both raw consumption and cooking, and – consumption of a daily handful of nuts.

Other recommendations were: three servings/day of skimmed dairy products, wholegrain cereals,

two-three servings of legumes/week, moderate to high consumption of fish, a low consumption of

 red and processed meat, avoidance of refined grains, processed baked goods, pre-sliced bread, soft

 drinks , fast foods and precooked meals. Emphasis was put on the use of EVOO for cooking and

dressings and the daily consumption of nuts. They were also recommended to walk 157 ≥30 minutes/day  and climbing stairs while possible.”

Please explain where this was taken from, what is a service, why proportions are not expressed as grams per day and what is a motivational lifestyle interview???

In essence a Mediterranean Style of diet would also be the preferable term.  Please consider.

You have not reported on the impact of walking 30 min per day nor recorded any data associated with this.  Please explain.

In the results tables you report the Medas Scores.  Was there any consideration given to the range in scores such as low medium and high to ascertain the potential outcomes???

Within the Nutritional intervention was the food supplied to the participants and why was a standardisation not used for servings as pieces make it difficult to determine the relationship to the outcomes.  Please explain.

In the results section you state: “HbA1c was significantly higher in women with GDM at 24-28 GW” and MEDAS and Nutrition scores were no significantly different between groups at 24-28 GW but were significantly higher in diabetic women at 36-38 GW” Please explain why this is in relation to the Med Diet style and what particularly is it about the diet that is resulting in these outcome.???

 In the last sentence of the manuscript you state: “On summary, this study translates evidence to clinical practice and provides a universal and easy to apply approach to reduce the burden of an increasing GDM incidence. In addition, an early nutritional intervention before GDM diagnosis is able to ameliorate differences in maternal-foetal adverse outcomes between diabetic and normoglycemic women.”  This may be the case for the data outcome, however, you would have enhance this substantially if you could please make a more positive statement for application into real clinical practise that says” Based on this research the identification of …………………with these identified attributes………. It is recommended that these women are encouraged to follow the following protocol to reduce the burden of GDM.

Ie diet requirement (grams per day), exercise requirement  ect………………………… a very positive contribution.

Author Response

Thank you very much for your kind comments and constructive suggestions. We agree with you in certain aspects and have provided a response to every issue raised. The changes applied are as follows:

Thank you for the opportunity to review your manuscript “Effectiveness of following Mediterranean diet recommendations in the real world in the incidence of gestational diabetes mellitus (GDM) and adverse maternal-foetal outcomes: a prospective, universal, interventional study with a single group. The St Carlos Study.”

Although a very interesting outcome, I offer the following for your consideration.

You have stated that this is a Mediterranean Diet, however, your nutritional intervention does not indicate this ; “Participants had an individual motivational lifestyle interview with a dietician one week after inclusion in visit 1. Nutritional intervention was based on the five basic MedDiet recommendations: - two servings/day of vegetables, - three pieces of fruit a day - avoiding juices, even fresh, - exclusive use of EVOO for both raw consumption and cooking, and – consumption of a daily handful of nuts. Other recommendations were: three servings/day of skimmed dairy products, wholegrain cereals, two-three servings of legumes/week, moderate to high consumption of fish, a low consumption of red and processed meat, avoidance of refined grains, processed baked goods, pre-sliced bread, soft drinks , fast foods and precooked meals. Emphasis was put on the use of EVOO for cooking and dressings and the daily consumption of nuts. They were also recommended to walk 157 ≥30 minutes/day  and climbing stairs while possible.”

Please explain where this was taken from, what is a service, why proportions are not expressed as grams per day and what is a motivational lifestyle interview???

While the definition of the Mediterranean diet can vary within regions, there are foundations that define it. As was recently shown in a review of the literature of the Mediterranean diet, the dietary recommendations are based on a frequent consumption of olive oil, vegetables, fruits, nuts, whole-wheat cereals, legumes, nuts, fish, poultry, dairy and avoidance of sweets and red/processed meats (doi: 10.3390/nu7115459). Our recommendations, are based on the recommendations proposed  by the Mediterranean Diet Foundation Expert Group (doi:10.1017/S1368980011002515). Alcohol was not included in our recommendations because its consumption is unadvised in pregnancy.

We considered that a serving of: vegetables was of 200-300 g, fruit was different depending on the fruit (one small banana of 60g, one medium-sized orange/apple/peach of 150g), dairy was of one glass of milk (250 ml) or 2 yoghurts (125 g each) and legumes was of one quarter of the plate (100 g). However, in this paper proportions are not expressed as grams because the aim of the intervention was to provide easy-to-apply dietary recommendations. We believe that giving recommendations with specific grams are not practical. Instead, we gave them serving sizes that they can determine visually.  For example, instead of telling women to consume 25-30g of nuts, we told them to eat a handful; instead of telling them to consume 300 g of vegetables, we told them to include them in every meal, at least as side-dishes but preferably occupying half of the plate; or, instead of telling them to eat 50-60g of cereals (rice, pasta, etc) we told them that a quarter of their plate should be composed of these carbohydrates.

A “motivational lifestyle interview” is an informative session were women were provided dietary advice and lifestyle guidance to have a healthy pregnancy. We refer to it as “motivational” because in this session women were explained the health benefits of following the lifestyle recommendations provided for themselves and offspring. In addition, women were given several meal and snack ideas compatible with the dietary recommendations; we believe this facilitated their adherence to the Mediterranean dietary pattern. Thus, motivating them to comply with the recommendations provided. To explain this concept we have included in line 171: “Women were explained the potential health benefits of following the dietary advice for themselves and offspring. In addition, women were given several meal and snack ideas where they could include all the different food options. We believe this made it easier and more appealing to adhere to the Mediterranean dietary pattern”.

In essence a Mediterranean Style of diet would also be the preferable term.  Please consider.

In agreement with this, we have changed the term in the manuscript in lines 91, 164 and 369.

You have not reported on the impact of walking 30 min per day nor recorded any data associated with this.  Please explain.

While we did register level of physical activity throughout the pregnancy, it is true that we have not provided data in this paper. Our analysis showed that performance of physical activity was scarce in all women, sharing similarities between groups. Therefore, we think that physical activity did not add or interfere with our results.  To avoid misconception, we have agreed to remove this from the paper (line 174).

In the results tables you report the Medas Scores.  Was there any consideration given to the range in scores such as low medium and high to ascertain the potential outcomes???

This is a very interesting question. We have recently published a paper were different degrees of adherence to MedDiet-based food targets were associated with adverse outcomes. We found that there was an inverse association between the degree of adherence and the rate of adverse outcomes (doi: 10.3390/nu11010066.). Since this has already been published, we did not think that performing a similar analysis would be necessary. Moreover, the aim of the present study was to assess the effect of a nutritional intervention based on a Mediterranean style diet in a real world setting.

Within the Nutritional intervention was the food supplied to the participants and why was a standardisation not used for servings as pieces make it difficult to determine the relationship to the outcomes.  Please explain.

No. The food was not supplied. This was a real-world study (not a randomized controlled trial) and we wanted to reproduce real-life as much as possible.

Our recommendations did not specifically provide how many grams there were per serving (as explained previously). However, when diet was evaluated in the lifestyle evaluation visits we did take into account serving sizes. Women were asked to specify the quantities in each case. For example, if they were not consuming half a plate of vegetables (200g aprox.) in each meal, they were not given that point. Therefore, we do think that we can associate diet and outcomes since the dietary evaluation did take into account serving sizes to obtain the MEDAS (and nutrition) score.  

In the results section you state: “HbA1c was significantly higher in women with GDM at 24-28 GW” and MEDAS and Nutrition scores were no significantly different between groups at 24-28 GW but were significantly higher in diabetic women at 36-38 GW” Please explain why this is in relation to the Med Diet style and what particularly is it about the diet that is resulting in these outcome.???

Women with GDM have an increased insulin resistance as compared to non-GDM women. Therefore, since glycemic control is impaired, HbA1c levels are higher in these women at the moment of GDM diagnosis (24-28 GW). Our results show that at this moment, dietary patterns are similar between GDM and non-GDM women. At 36-38 gestational weeks, three months after GDM diagnosis and treatment (73% diet-controlled), these women had significantly better dietary habits (as shown by the Nutrition and MEDAS score) and similar HbaA1c levels as non-GDM women. Women with GDM have an increased insulin resistance. A Mediterranean diet has been documented to be successful in improving insulin sensitivity and in the treatment of type 2 diabetes. Thus, we suspect that this improvement in HbA1c levels (i.e. glycemic control) can be attributed to the higher adherence to the Mediterranean diet. This dietary pattern is known for having a high antioxidant and anti-inflammatory capacity, both of which exhibit antidiabetic properties. This concept is explained in lines 355-372.

In the last sentence of the manuscript you state: “On summary, this study translates evidence to clinical practice and provides a universal and easy to apply approach to reduce the burden of an increasing GDM incidence. In addition, an early nutritional intervention before GDM diagnosis is able to ameliorate differences in maternal-foetal adverse outcomes between diabetic and normoglycemic women.”  This may be the case for the data outcome, however, you would have enhance this substantially if you could please make a more positive statement for application into real clinical practise that says” Based on this research the identification of …………………with these identified attributes………. It is recommended that these women are encouraged to follow the following protocol to reduce the burden of GDM.

We have agreed to rephrase the last sentence of the manuscript following you suggestion:  “On summary, based on the translation of evidence to clinical practice we recommend that all pregnant women are encouraged to follow this protocol from early in pregnancy to reduce the burden of GDM” (lines 391-393).

Reviewer 2 Report

Thank you for allowing me to review this manuscript of the MedDiet for GDM risk. The authors correctly describe the need to identify effective dietary interventions for pregnant women to prevent GDM. Based on their results from a previous MedDiet trial, the authors conducted another, without providing the olive oil and nuts, in attempt to assess implementation in a more real world setting. The objective is interesting. However, there are significant concerns about the design, and fatal flaws in the interpretation of the methods.

The registered clinical trial website states this is non-randomized, and the description of the methods also suggests this isn’t randomized. Yet the authors state this is a RCT.

Further, the abstract is confusing. It states that a RCT was conducted, but does not describe the design or interventions being compared. The results are presented as differences between GDM and non-GDM women, but conclusions state that Med Diet was associated with a number of outcomes.

Given the study design, which seems to be a single arm intervention, it is not possible to draw conclusions about the efficacy of the intervention, without an appropriate comparator group. Further, the results expand upon the differences between GDM and non-GDM women, without discussion of the Med Diet, which does not seem to be consistent with the aim of this study. Significant rework of the manuscript is suggested. 

Author Response

Thank you very much for your suggestions with which we agree mostly. Therefore we have provided an answer to every comment that we hope you find appropriate.

We considered Registered Clinical Trial (RCT) after the ICMJE definition of clinical trials. In June 2007 the ICMJE adopted the WHO's definition of clinical trial: "any research study that prospectively assigns human participants or groups of humans to one or more health-related interventions to evaluate the effects on health outcomes." Health-related interventions include any intervention used to modify a biomedical or health-related outcome (for example, drugs, surgical procedures, devices, behavioral treatments, dietary interventions, and process-of-care changes). Health outcomes include any biomedical or health-related measures obtained in patients or participants, including pharmacokinetic measures and adverse events.

It does not necessarily have to be randomized.

However, we agree that this definition of RCT (Registered Clinical Trial instead of Randomized Clinical Trial) can lead to confusion, and therefore, following your comments we have removed RCT from
the abstract (line 31). Instead we have stated ¨This trial is a prospective, clinic-based, interventional study with a single group”.

The results of this study are presented primarily as differences in GDM rate between an easy, universal nutritional intervention in the real world based on a MedDiet, and a previously reported randomized clinical trial were EVOO and nuts were provided to ensure adherence.  Regarding maternal-foetal outcomes, we do not pretend to draw conclusions about the efficacy of the nutritional intervention, since this was already shown in the previous RCT (14). In this randomized study women in the intervention group had significantly reduced rates of insulin-treated GDM, prematurity, gestational weight gain, emergency C-sections, perineal trauma, and SGA and LGA new-borns compared to the control group.  Given this improvement in several maternal and neonatal outcomes with a MedDiet we have assessed in the present study whether perinatal adverse outcomes, known to be associated to GDM, could be similar between women with GDM and normal glucose tolerance when the nutritional intervention is applied universally and early in pregnancy.

Nevertheless, we agree that the description of the objectives and results could benefit from some clarification. Accordingly we have deleted from the abstract: “In order to assess its translational effects in the real world, we evaluated the effect of MedDiet from 1st gestational visit in GDM rate and maternal-foetal outcomes “(lines 29-31). Instead we have included: “In order to assess its translational effects in the real world we evaluated the effect of MedDiet from 1st gestational visit in GDM rate compared with control (CG) and intervention (IG) groups from the previously referred trial. As secondary objective we also compared adverse perinatal outcomes between normoglycemic and diabetic women”. In addition we have included in line 39: “GDM was diagnosed in 13.9%. This rate was significantly lower than the CG: RR 0.81 (0.73-9.28), p<0.001 and no different from the IG: RR 0.96 (0.85-1.07), p=0.468”.

We have included the according changes in the main text:

·        Research Design and Methods: we have substituted lines 93-102 by the following paragraph: “Our primary objective is to evaluate in women with normal fasting blood glucose [FBG<92 mg/dL (5.1 mmol/L)] at the first gestational visit (8-12 gestational weeks [GW]) the effect of Mediterranean-based nutritional recommendations in the real world in the incidence of GDM and assess its effectiveness compared with the control group from the previous clinical trial (14).

Secondary outcome measures are: to compare the incidence of GDM with the MedDiet intervention in the real world versus the intervention group from the previous clinical trial (14), and to assess the effect of the dietary intervention on the percent of diabetic women requiring insulin therapy, gestational weight gain (GWG) , pregnancy-induced hypertension, caesarean section , perineal trauma, preterm delivery (< 37 GW), neonates SGA (<10 percentile) and LGA ( >90 percentile) according to national charts, admissions to the Neonatal Intensive Care Unit (NICU), and UTIs.  We will compare adverse perinatal outcomes in women diagnosed with GDM and women with NGT, since an early nutritional intervention previous to GDM diagnosis could ameliorate differences between groups”.

·        Results: we have included in line 246: “GDM was diagnosed in 130 women (13.9%).  This incidence was significantly lower than in the control group: RR 0.81 (0.727-9.28), p<0.001 and no different from the intervention group: RR 0.96 (0.85-1.07), p=0.468 from the St. Carlos GDM prevention study”. 

Round 2

Reviewer 1 Report

Dear Authors,

Thank you for the opportunity to assess your revised manuscript “Effectiveness of following Mediterranean diet recommendations in the real world in the incidence of gestational diabetes mellitus (GDM) and adverse maternal-foetal outcomes: a prospective, universal, interventional study with a single group. The St Carlos Study.”

Thank you for addressing what I can identify as changes, however, a track changes copy of the manuscript would have been very helpful.

I also note that you have not endevoured to explain why the Med Diet has given you these results.  What is it about the diet assessment you have made that has resulted in the outcomes observed? This form of explanation, would potentially give this manuscript greater readability and foundation for further insights.

Author Response

Thank you very much for your kind comments. With regards to the tracked-changes copy, we are sorry that you could not have access to this version. There must have been a problem with the upload of this file. We are going to make sure that you get access to the tracked-changes version this time. Our apologies.  

In addition, we agree that some explanation should be added as to why the MedDiet could improve these outcomes. To support our statements, we have added two references to the manuscript (28,29).

The changes applied are as follows:

Dear Authors,

Thank you for the opportunity to assess your revised manuscript “Effectiveness of following Mediterranean diet recommendations in the real world in the incidence of gestational diabetes mellitus (GDM) and adverse maternal-foetal outcomes: a prospective, universal, interventional study with a single group. The St Carlos Study.”

Thank you for addressing what I can identify as changes, however, a track changes copy of the manuscript would have been very helpful.

I also note that you have not endevoured to explain why the Med Diet has given you these results.  What is it about the diet assessment you have made that has resulted in the outcomes observed? This form of explanation, would potentially give this manuscript greater readability and foundation for further insights.

We have added a few sentences in lines 370-377 (in the tracked-changes version):

“Pregnancy complications such as GDM, prematurity, UTI, new-born birthweight (large- and small-for- gestational- age) and pregnancy-induced hypertensive disorders have been associated with maternal diet. All these complications seem to be associated with proiinflammatory processes and/or impaired glycaemic control. Dietary patterns rich in vegetables, fruits, fibre, fish, legumes, whole-wheat cereals and vegetable oil have shown to favour these complications (28,29).  The anti-inflammatory and antioxidant properties of this diet, along with its shown effects on improved insulin sensitivity could explain these results”.

To support this statement we have added two references to the manuscript (lines 500-505 tracked-changes version):

28. Chen X, Zhao D, Mao X, Xia Y, Baker PN, Zhang H. Maternal Dietary Patterns and Pregnancy Outcome. Nutrients. 2016;8. doi:10.3390/nu8060351

29. Kibret KT, Chojenta C, Gresham E, Tegegne TK, Loxton D. Maternal dietary patterns and risk of adverse pregnancy (hypertensive disorders of pregnancy and gestational diabetes mellitus) and birth (preterm birth and low birth weight) outcomes: a systematic review and meta-analysis. Public Health Nutr. 2019;22:506-520.
